

# Foundational models and federated learning: survey, taxonomy, challenges and practical insights

Cosmin-Andrei Hatfaludi[1,2] and Alex Serban[1,2]

[1] Foundational Technologies, Siemens SRL, Brasov, Romania
[2] Automation and Information Technology, Transilvania University, Brasov, Romania

## ABSTRACT

Federated learning has the potential to unlock siloed data and distributed resources by enabling collaborative model training without sharing private data. As more complex foundational models gain widespread use, the need to expand training resources and integrate privately owned data grows as well. In this article, we explore the intersection of federated learning and foundational models, aiming to identify, categorize, and characterize technical methods that integrate the two paradigms. As a unified survey is currently unavailable, we present a literature survey structured around a novel taxonomy that follows the development life-cycle stages, along with a technical comparison of available methods. Additionally, we provide practical insights and guidelines for implementing and evolving these methods, with a specific focus on the healthcare domain as a case study, where the potential impact of federated learning and foundational models is considered significant. Our survey covers multiple intersecting topics, including but not limited to federated learning, self-supervised learning, fine-tuning, distillation, and transfer learning. Initially, we retrieved and reviewed a set of over 4,200 articles. This collection was narrowed to more than 250 thoroughly reviewed articles through inclusion criteria, featuring 42 unique methods. The methods were used to construct the taxonomy and enabled their comparison based on complexity, efficiency, and scalability. We present these results as a self-contained overview that not only summarizes the state of the field but also provides insights into the practical aspects of adopting, evolving, and integrating foundational models with federated learning.

## INTRODUCTION

Federated learning (FL) allows multiple parties to collaboratively train machine learning (ML) models without exchanging or transferring private data (*Zhang et al., 2021*). Its main goal is to improve ML algorithms by integrating siloed data, which would otherwise remain isolated and underused. For example, using patient data from different hospitals to improve diagnostic ML algorithms would be challenging without FL (*Pfitzner, Steckhan & Arnrich, 2021*).

Corresponding author
Cosmin-Andrei Hatfaludi,
cosmin.hatfaludi@siemens.com

As ML models and datasets grow, particularly with the development of foundational models (FMs) (*Bommasani et al., 2021*), training them requires more computational resources and diverse data. However, high-quality data is often siloed due to privacy concerns. In this context, it is valuable to explore how FL can enable new data sources for training FMs, improve resource sharing, or use pre-trained FMs, to develop robust collaborative ML models. While both FMs and FL aim to improve ML models, their approaches differ fundamentally. FMs are trained centrally using large and diverse datasets, and fine-tuned for various tasks (*Awais et al., 2025*; *Chang et al., 2024*; *Liang et al., 2024*). In contrast, FL allows multiple parties to train specialized models without centralizing data, preserving privacy. Their integration faces technical challenges such as bandwidth management, training latency, and efficient data transmission (*Zhang & Metaxas, 2023*), as well as legal and organizational challenges such as finding incentives to defining Internet Protocol (IP) ownership for participating nodes (*Woisetschläger et al., 2024*).

Navigating the intersection of FMs and FL remains challenging due to the absence of comprehensive reviews. While existing literature addresses the challenges of integrating FMs and FL (*Zhuang, Chen & Lyu, 2023*), examines their integration at various development stages (*Woisetschläger et al., 2024*; *Kang et al., 2023*), and categorizes existing literature (*Ren et al., 2024*; *Li et al., 2024b*), an integrated an self-contained review is still missing. For example, *Zhuang, Chen & Lyu (2023)* lack a structured taxonomy or technical analysis of methods, *Ren et al. (2024)* and *Li et al. (2024b)* propose taxonomies, but many of their categories overlap, making it hard to uniquely position new methods, while *Woisetschläger et al. (2024)* and *Kang et al. (2023)* focus on specific stages such as training.

To address these gaps, we aim to provide a more comprehensive survey structured around a non-overlapping taxonomy that follows the development life-cycle stages. Furthermore, we aim to provide a comprehensive technical comparison of the available methods, together with practical insights into their implementation and quality attributes, and study their application in specific use-cases. This will guide future researchers and practitioners understand, apply, and evolve these methods.

Overall, the contributions of this article are as follows:

- We present a comprehensive literature survey that aims to cover the majority of articles at the intersection of FMs and FL.
- We categorize the articles using a novel taxonomy based on the stage where FM are used (*e.g.*, pretrain or inference) and the methods used at each stage.
- We compare the methods based on complexity, efficiency and scalability.
- We discuss practical aspects of applying the methods presented and provide guidelines and future research directions, focusing on the application of FL and FMs in the healthcare domain.

The remainder of this article is organised as follows. We start with background information and related work (Background and Related Work), followed by the methodology and the taxonomy used for the literature survey (Survey Methodology). We then discuss the selected articles (Methods) and practical aspects of the methods (Practical

perspectives). The article concludes with a discussion of study limitations (Limitations), findings (Discussion), followed by conclusions (Conclusions). Prerequisite information about FL, FMs, fine-tuning, and KD is provided in the Appendix.

## BACKGROUND AND RELATED WORK

Several comprehensive surveys have been published covering FL or FMs, addressing applications in multiple domains including healthcare. For example, *Zhang et al. (2021)* present a survey of FL, categorizing and summarizing various methods along with their advantages and disadvantages. *Liu et al. (2024)* build upon this survey, incorporating more recent techniques and providing an extensive taxonomy for their classification. Additionally, several articles explore the challenges and opportunities in FL—*e.g.*, (*Wen et al., 2023*)—and examine different quality attributes of the algorithms, such as trustworthiness (*Zhang et al., 2024b*), robustness (*Huang et al., 2024*), and fairness (*Ji et al., 2024*). In the medical domain, *Guan et al. (2024)* and *Pfitzner, Steckhan & Arnrich (2021)* cover both the techniques used as well as benchmarks and datasets for FL.

Likewise, numerous surveys address FMs for various applications, including vision (*Awais et al., 2025*), language (*Chang et al., 2024*), and multi-modal FMs (*Liang et al., 2024*). These surveys include methods for developing FMs, comparative benchmarks, as well as the opportunities and risks associated with using FMs (*Bommasani et al., 2021*). In the medical domain, *Azad et al. (2023)* present a comprehensive survey discussing algorithms, modalities, and various organs for which FMs have been developed. Within this realm, FMs are also called generalist models (*Moor et al., 2023*). *Zhang & Metaxas (2023)* present a classification of the types of FMs and generalist models, ranging from very broad, multi-modal, models, to very narrow, task-specific FMs.

Nonetheless, the majority of these studies concentrate only on the development and scaling of FMs or FL algorithms independently. Although several surveys discuss specific algorithms applicable to both FL and FMs, such as knowledge distillation (KD) (*Wu et al., 2022*)—they do not provide a comprehensive understanding of the intersection between FL and FMs (*Chen et al., 2023*; *Li & Wang, 2024*; *Yu, Muñoz & Jannesari, 2023*).

While the intersection of FL and FMs is still developing, a series of articles highlight the opportunities and challenges of integrating these fields, as well as partial surveys and taxonomies. *Zhuang, Chen & Lyu (2023)* provide an overview of the motivations and challenges associated with using FMs in FL. While some of the authors' motivations are realistic—such as overcoming a shortage of available private or personalised data for developing FMs—are less so. For example, attempting to address response delays or service downtimes for FMs by running them in FL may be impractical, as FL introduces additional communication overhead that can increase latency and provide additional disruptions in service. We observe that all the presented challenges—such as those related to reducing communication and computational costs, or developing incentive mechanisms for collaborations—are timely, and are also acknowledged in similar position papers exploring the interplay between FL and FMs.

Several related surveys focus on specific aspects of FL and FMs. For example, *Woisetschläger et al. (2024)* present a survey of methods for training FMs in FL settings,

while *Kang et al. (2023)* survey methods for using transfer learning in FL. While these studies are closely related, our aim is to provide a broader and more comprehensive technical survey that will encompass specific concerns as presented in these works.

The studies by *Ren et al. (2024)* and *Li et al. (2024b)* come closest to our aim, as they seek to provide taxonomies and surveys of the intersection of FL and FMs. When compared with *Ren et al. (2024)*, our objectives are slightly divergent. In our study, we focus only on technical methods, which broadly fall under the categories of efficiency and trustworthiness from their taxonomy, and avoid broader debates on topics such as incentivization or alignment. Furthermore, our aim is to discuss these methods, rather than characterize and classify them in general terms, or present their motivation. This approach allows us to present a more comprehensive and self-contained technical survey, enabling readers to more effectively understand, apply, and evolve the methods presented.

Moreover, we develop a more focused, technical, taxonomy of the methods presented, derived from the underlying principles used to develop them, and ensuring there are no overlaps between the selected classes. For example, *Ren et al. (2024)* propose different classes such as computational and communication efficiency, although most methods for computational efficiency also improve communication efficiency. Instead, we identify orthogonal criteria for classifying the methods presented. Additionally, we avoid speculative debates, such as whether quantum computers can aid FL and FMs, as these technologies are not considered viable in the near future. In this sense, our study is closer to the work of *Li et al. (2024b)*, but instead of focusing on representative methods, we aim to present potentially all available methods, and provide a more detailed and structured characterisation of the methods presented. We also provide in-depth details on practical application and implementations of FMs with FL in the medical domain. We highlight these differences in Table 1.

## SURVEY METHODOLOGY

In the following sections, we present the methodology used to conduct the literature survey and develop the taxonomy used throughout the article.

### Study design and protocol

Our study can be classified as a literature survey (*Sataloff et al., 2021*), aimed at identifying, classifying, and evaluating the strengths and weaknesses of various methods proposed at the intersection of FL and FMs. Towards this goal, we designed the study protocol following the guidelines of *Okoli (2015)* and *Yannascoli et al. (2013)*.

A total of five research questions, as summarized in Table 2, were formulated to guide the study. The questions cover a range of topics, from the exploration of general methods to more practical aspects of implementation, and future research directions. Specifically, RQ1 and RQ2 focus on the general intersection of FL and FMs, aiming to identify and classify the key methods in this area. RQ3 and RQ4 focus on practical aspects, investigating the applications and best practices for implementing these methods in real-world scenarios, particularly in the medical domain. RQ5 explores future directions and open questions, providing insights into future research.

**Table 1** Comparison with prior surveys across key dimensions: the structure of the taxonomy (including overlaps among subclasses), the extent of lifecycle coverage (spanning training, customization, and deployment), the depth of technical comparison (such as method comparisons or ranking), the inclusion of practical insights (addressing real-world considerations), and the analysis of use cases (evaluating methods in applied contexts).

| Study | Non-overlapping Taxonomy | Lifecycle coverage | Technical comparison | Practical Insights | Use-case analysis |
|---|---|---|---|---|---|
| *Zhuang, Chen & Lyu (2023)* | ✕ | ✕ | ✕ | ✕ | ✕ |
| *Ren et al. (2024)* | ✕ | ✕ | ✓ | ✕ | ✕ |
| *Li et al. (2024b)* | ✕ | ✕ | ✓ | ✕ | ✕ |
| *Woisetschläger et al. (2024)* | ✕ | ✕ | ✕ | ✕ | ✕ |
| *Kang et al. (2023)* | ✕ | ✕ | ✕ | ✕ | ✕ |
| **This article** | ✓ | ✓ | ✓ | ✓ | ✓ |

**Table 2** Research questions that guided this study, as outlined in our study protocol.

| ID | Research Question |
|---|---|
| RQ1 | What are the techniques that integrate FMs with FL? |
| RQ2 | What are the trade-offs involved in integrating FMs with FL? |
| RQ3 | What are the most practical methods for using FMs with FL? |
| RQ4 | What FL and FMs methods have been applied in the medical field? |
| RQ5 | What are the key open research questions and future directions for developing FMs with FL? |

To ensure comprehensive coverage of academic and potential non-academic studies, the search strategy included multiple information sources. Initially, we used the Google and Google Scholar search engines. While these engines are known to encompass many other information sources commonly used in literature surveys (*Shahin, Babar & Zhu, 2017*), we also used the ScienceDirect and Scopus search engines to further broaden the search. For each information source we used the first five pages of answers. ScienceDirect and Scopus were queried using the APIs, limiting the number of answers to fifty for each query.

To define the queries, we followed several guidelines (*Okoli, 2015*; *Yannascoli et al., 2013*), and initially defined terms that comprised of two elements. For the first element, we used the term *federated learning*, and for the second element we used the term *foundational model* along with various synonyms for the techniques used to develop or use FMs. These included *self-supervised learning*, *pre-trained models*, *fine-tuning*, *distillation*, *transfer learning*. Additionally, we created two variations of these queries to make them more specific. First, we added an element suggesting the medical domain using the words *medical* and *healthcare*. Second, we added an element suggesting the practical application of these methods using the words *application*, *implementation*, *development*, and *deployment*. A complete list of keywords is provided in the Appendix.

To filter out irrelevant documents, we first removed duplicates based on the article titles, and restricted the search to articles published from 2021 onwards, as this year

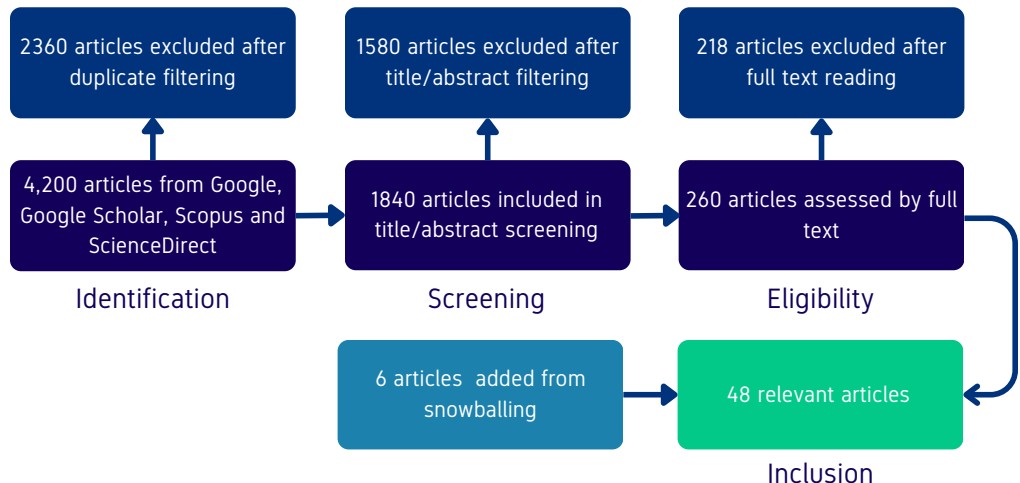

**Figure 1** PRISMA flow chart illustrating the article selection process at each stage of the filtering criteria, as defined in the study protocol.

corresponds with the first developments in FMs. Afterwards, we reviewed the titles and abstracts of the remaining articles and applied several inclusion and exclusion criteria. Inclusion criteria required that articles (i) addressed at least one of the RQs, (ii) were peer-reviewed or appeared in more reputable preprint servers, and (iii) were written in English. Exclusion criteria eliminated articles that (i) were not accessible in full text, (ii) focused solely on unrelated domains, or (iii) appeared to lack a clear methodological contribution. The remaining articles were fully read and evaluated using these criteria, and the initial list of articles was complemented from their references using a snowballing strategy (_Jalali & Wohlin, 2012_). An illustration of the article filtering process is provided in Fig. 1.

## Taxonomy

When attempting to classify the articles discussed in this study using previously introduced taxonomies, we found that no existing work could categorize them into distinct, non-overlapping classes based on the algorithms presented. For example, the taxonomy proposed by _Li et al. (2024b)_, while compelling, exhibited significant overlap between classes and provided only representative examples for each technique. Upon trying to scale up their taxonomy, we found that the fine granularity of their approach made it difficult to fit some methods from our study. Therefore, we developed a broader taxonomy that is compatible with all proposals in the literature but adheres to more stringent criteria.

Initially, we classified the articles based on the stage of the development life-cycle into three categories: (i) _train_—focusing on techniques for pre-training FMs using FL, (ii) _customize_—focusing on the adaptation of FMs for specific tasks using FL, and (iii) _deploy_—focusing on the use of FL to run inference for a pre-trained or customised FM. Within each of these categories, we further defined sub-classes based on the core algorithmic technique used to develop the methods. An illustration of this taxonomy is provided in

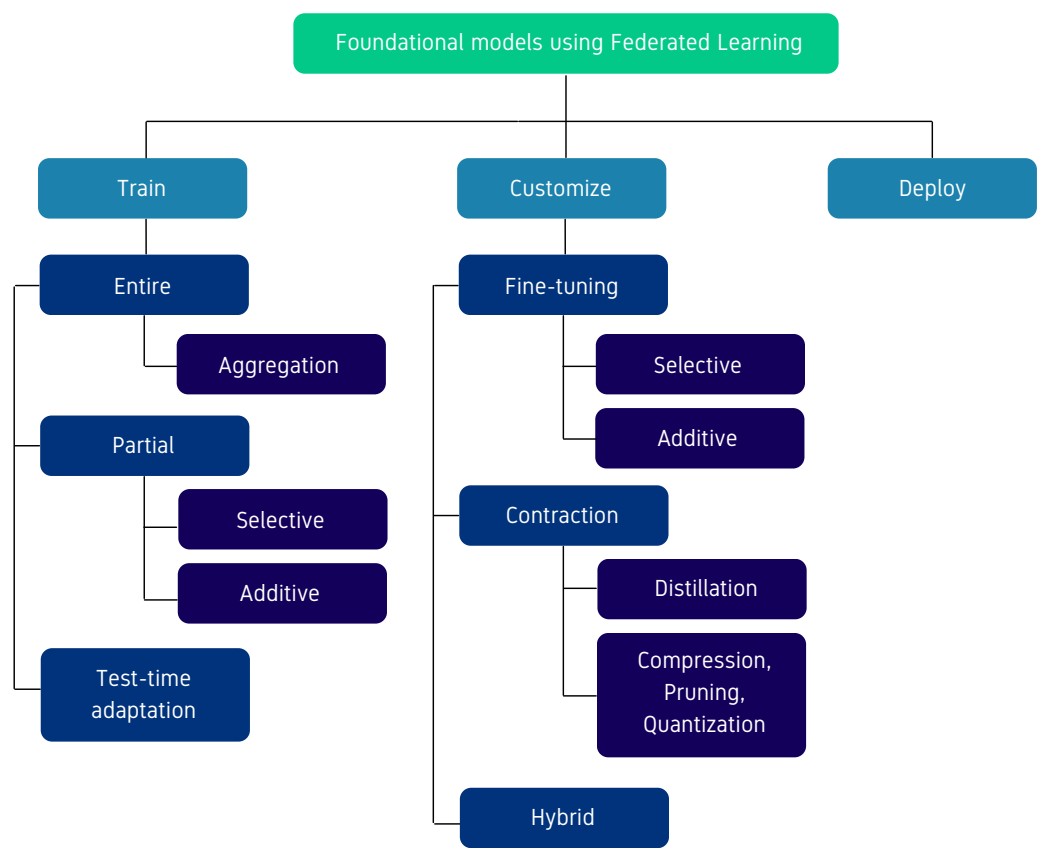

**Figure 2** Taxonomy of methods for training, customizing or deploying foundational models using federating learning.

Fig. 2. We observe that for certain classes, such as those used to customize FMs, it was possible to define more fine-grained sub-classes due to the diversity of the algorithms used. Conversely, for other classes like the deploy class, were fewer techniques have been developed, the taxonomy is more coarse. Comprehensive details about the class definitions are discussed in Methods, where the methods belonging to each class are introduced.

To ensure that the validity of the taxonomy, we conducted an internal validation with experts within our team. Two experts independently assigned each method to one of the taxonomy classes, followed by a review to resolve any disagreements and ensure consistency. Furthermore, cross-referencing of prior taxonomies and internal discussions between the authors and the experts helped reached consensus. Both experts were researchers with over five years of experience in machine learning and at least three years of focused research in FL or FMs.

## METHODS

This section presents and compares the methods identified based on the taxonomy from Fig. 2. An illustration of the methods grouping by year is illustrated in Fig. 3, following the three main categories of the taxonomy. We can observe a

slight shift from methods focusing on training the FM in FL to methods focusing mostly on customization. This is likely due to the increase in resources needed to train FMs over time and the emergence of more efficient fine-tuning methods such as low-rank adaptation (LoRA).

## Train

Although collaboratively training FMs using FL can offer access to high-quality siloed data, it is also the most challenging use-case as communicating large model updates leads to substantial communication overhead. Managing large datasets and models requires innovative communication-efficient algorithms and compression methods (*McMahan et al., 2017*). Additionally, handling heterogeneous data distributions and varied computational resources across nodes poses further challenges (*Kairouz et al., 2021*). Due to these challenges, few publications attempt to pre-train entire FMs, with most focusing on adding parameters or pre-training only a subset of the FMs collaboratively. This technique is also known as continuous domain adaptation, where off-the-shelf FMs are further pre-trained on more specific data (*Jiang, Svoboda & Lane, 2023*).

To initially categorize the algorithms in this class, we consider the parts of the model used for pre-training: (i) *entire*—where the whole model is pre-trained using FL, (ii) *partial*—where only parts of the model are pre-trained using FL, and (iii) *test time adaptation*—where the pre-training phase adjusts specific parameters or inputs for customization or deployment using FL—such as modifying the prompts used to better suit the context or specific user requirements. An illustration of these methods is provided in Fig. 4, focusing on the parts of the models that are trained collaboratively. On the server side, most algorithms discussed in this section use Federated Averaging (FedAVG) (details in Appendix) or its variants for parameter aggregation, but implement novel techniques to minimize the parameters transmitted over the network. Within these classes, several sub-categories could be identified, detailed in the next sections.

### *Entire model training*

The methods presented in this section can be classified into methods that aim to pre-train FMs using FL, discussed below, and those that improve the aggregation algorithms used on the server, detailed in the *aggregation* paragraph. This distinction was made because novel aggregation methods can, in principle, be combined with other pre-training techniques.

*Garcia Bernal et al. (2021)* conducted one of the first empirical studies on pre-training a FM in FL, training a Word2Vec model (*Mikolov et al., 2013*) collaboratively with a small number of clients holding large text corpora. They used FederatedSGD (*McMahan et al., 2017*), where the only changes were context-specific decisions like merging client vocabularies. While the final model didn't improve performance significantly, the study showed the feasibility of training FMs and FL. *Sani et al. (2024)* further optimized training by combining efficient local and global gradient updates with data parallelism techniques typically used for large-scale FMs (*e.g.*, distributed data parallelism). Besides these optimizations, the authors used a larger set of local updates before updating the global model, which reduced communication costs and proved effective when scaling the dataset size.

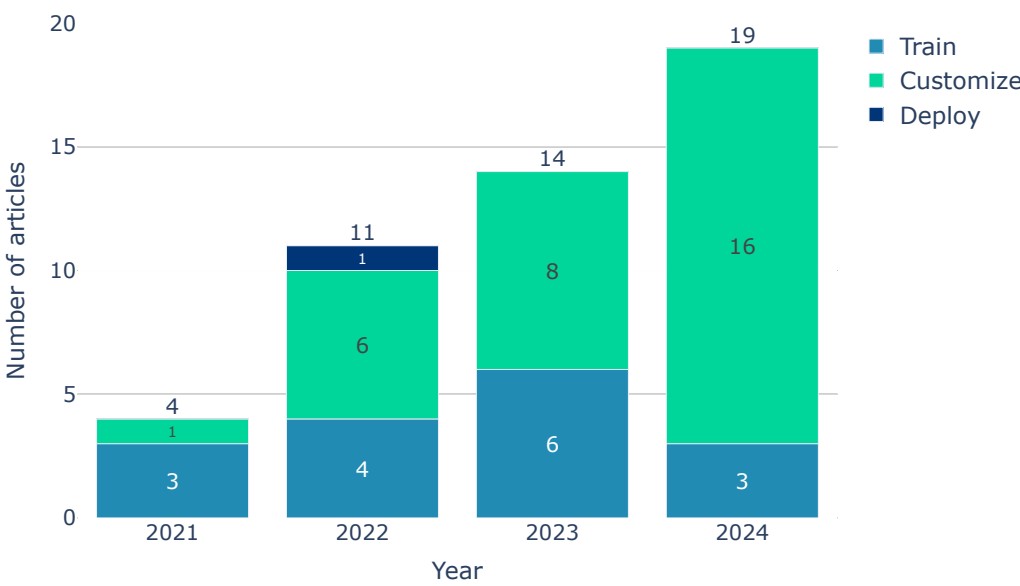

**Figure 3** Temporal trends in the evolution of methods over the years following the taxonomy illustrated in **Fig. 2**.

While these articles focused on using existing techniques without introducing novel methods, they demonstrate the viability of pre-training FMs in FL settings. *Zhuang et al. (2021)* first adapted pre-training techniques for FMs by modifying Bootstrap Your Own Latent (BYOL) (details in Appendix) to run in FL. In their approach, each client trains a BYOL-like model locally using contrastive learning, with one encoder updated *via* gradient descent and the other *via* exponential moving average (EMA). Every *n* epochs, clients upload their EMA encoders to the server, which merges them using FedAVG and sends the updates back to the clients. The clients can then dynamically choose to replace their EMA models with the global one while maintaining consistency with their local data. The novelty of this approach is that only the EMA encoder is updated globally, reducing communication overhead. *Li, He & Song (2021)* also introduced a method using contrastive learning in FL, allowing clients to adapt their local models on demand. Unlike *Zhuang et al. (2021)*, they performed contrastive learning at the model level, aiming to minimize the distance between local and global model features by contrasting them at each training step. The global model is updated using FedAVG every epoch, and after the update the contrastive loss is used to update the local model. This removes the need for an EMA encoder, as the global model acts like it. Nevertheless, the communication costs increase as the global model is updated at every epoch.

*Zhuang, Wen & Zhang (2022)* examined the advantages and disadvantages of choosing different pre-training techniques, and evaluated their impact of running them in FL. They conducted an empirical study comparing SimCLR, MoCo, BYOL, and SimSiam (details in Appendix) pre-training methods in FL. The authors found that non-contrastive methods such as BYOL perform better, and while EMA is not essential, it improves performance. To further investigate EMA's influence, they replaced the update of the local EMA encoder
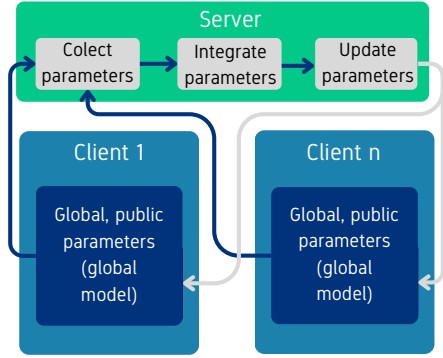

**(a)** Entire foundational models' training using federated learning, where all model parameters are adjusted.

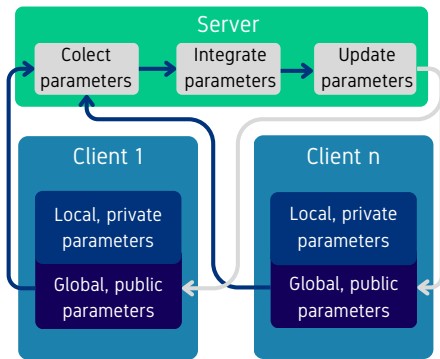

**(b)** Partial foundational models' training using federated learning, where only a subset of global parameters are adjusted.

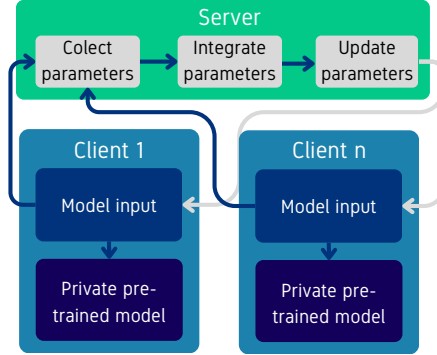

**(c)** Test time adaptation of foundational model using federated learning, where only the models' input is adjusted.

**Figure 4** **Overview of pre-training methods for foundational models using federated learning.**

with the global model, as performed in *Zhuang et al. (2021)*, with an EMA update on the global model where the decay rate is dynamically set by all clients. This technique led to performance improvements, as the EMA model is more adapted to local data.

*Makhija, Ho & Ghosh (2022)* presented a method where each client first pre-trains an individual model, and then a global model is adapted using an independent alignment dataset. At each epoch, the server sends a subset of this dataset and the corresponding global model embeddings to all clients. Clients then compute the similarity between their local embeddings and the global embeddings, and send their local embeddings back to the server. The server aggregates these representations and sends the results back to the clients, which continue training by maximizing the embeddings similarity. While this approach outperforms previously discussed methods such as *Zhuang, Wen & Zhang (2022)*, it incurs high communication costs due to the exchange of data and embeddings. Some of these costs are alleviated by using a smaller alignment dataset. However, the need for a small dataset is not guaranteed by the method. *Agarwal, Rezagholizadeh & Parthasarathi (2023)* ran an empirical study on pre-training FMs using FL, investigating distinct factors such as

data heterogeneity, the clients contribution to the final metrics, as well as the influence of the client's dataset size in relation to performance and speed. They found that using FMs can have both a positive and a negative impact on the local models, depending on how skewed the data distribution is between clients. The authors also found that having more clients with less data, instead of less clients with more data, can impact the performance of the FMs, and note that performing less updates from clients with less data can improve the communication efficiency without loss of performance.

*Aggregation.*  To improve FedAVG and address potential scaling issues with the embedding norms during local updates, *Kim et al. (2023)* introduced an additional L2 feature normalization, moderated by a client-specific scaling factor. This technique demonstrates improved performance when applied to both pre-training and fine-tuning, and could potentially be integrated with the methods discussed earlier.

*Rehman et al. (2023)* introduced a new aggregation method aimed at reducing bias from heterogeneous clients during model aggregation. In their approach, each client performs local pre-training and uploads its models to the server. The server then computes a layer-wise weight for each model's contribution to the final global model, using a weight metric measured as the cosine similarity between the client's model and the global model from the previous iteration. The method can be interpreted as a weighted, layer-wise FedAVG, which improves both the performance and mitigates some of the biases introduced by clients with more skewed data distributions.

*Recasens et al. (2023)* proposed a novel averaging technique based on the Fisher matrix, which merges local models in the parameter space. The method assumes that each client performs pre-training locally on all data, until convergence, and only upon convergence merges the models into a global model, significantly reducing the communication costs during training. The Fisher matrix is used in the aggregation process to weigh the importance of each parameter, providing information for preserving parameters that contribute most significantly to the model's performance. This method is particularly effective in scenarios where client data distributions are highly heterogeneous, ensuring that the global model benefits from the most informative local updates.

### Partial model training

The methods discussed in this section pre-train only a portion of the FMs using FL and can be classified based on the parameters used for pre-training. These include methods that add new parameters for pre-training (*additive*) and methods that select existing parameters for pre-training (*selective*).

*Additive.*  *Tan et al. (2022)* assume that the clients already have independent pre-trained models, obtained either from pre-training on their own data or adopting off-the-shelf FMs, and use these to pre-train a global FM. In this setting, each client encodes their data with the local model and shares the embeddings with all other clients. The clients then concatenate all embeddings and project them through a learnable projection layer to obtain client-specific prototypes for the concatenated embeddings. These prototypes are shared with the server, which aggregates them using FedAVG, and sends the results

back to the clients. For supervised or semi-supervised tasks, multiple class-wise prototypes can be used. After receiving the prototypes from the server, each client minimizes the distance between the local and the global prototype, or can perform contrastive learning if multiple prototypes are used. Sharing and training only the prototypes significantly reduces communication costs while enabling each client to contribute more effectively to the global model. This method improves both performance and communication efficiency. However, it assumes the existence of pre-trained models for all clients and does not explore the scenario where all clients use the same FM (*e.g.*, using an off-the-shelf FM). *Fanì et al. (2023)* extended this method by introducing a regularization term based on the magnitude of the prototype weights to address concerns arising from heterogeneous data distributions across clients. This addition resulted in further performance improvements.

*Chen, Su & Li (2024)* further used the prototype mechanism to achieve a balance between pre-trained models that provide generic features and more personalized, client-specific features. The adaptation consists in increasing the training period for clients with more complex data, and simplifying the local parameters and the training period for clients with limited or noisy data. This dynamic adjustment helps optimize the training process for each client. Their experiments demonstrated improved performance in pre-training methods for personalizing the final models, while being more resilient to bias added by specific clients.

*Kim, Yoo & Kang (2023)* further simplified this method demonstrating that removing the embedding concatenation and projecting only the features from the local models is also effective. The clients send only the projection layers to the server, which aggregates them using FedAVG. However, the authors used projection layers with many more parameters, making them more similar to adapter layers (*Pfeiffer et al., 2021*) rather than simple projection layers.

*Lu et al. (2024)* further extended this strategy by also incorporating learnable adapters for the inputs, in addition to the projections (prototypes) used for the outputs of a FM. The authors trained both an auto-encoder to map the inputs to a FM (a stage also called input surgery), and use the outputs of the FM to project them to a common space through a learnable layer. The server receives both the input auto-encoder and the output projection, performs FedAVG, and returns the aggregated results to the clients, who then replace their corresponding local components with the global ones. This approach improves the adaptability of the FM by integrating both input and output adjustments. However, it also increases communication costs compared with previously mentioned methods, as the input auto-encoder is additionally trained using FL.

*Selective.* *Lit et al. (2022)* introduced a method to pre-train the BERT language model in FL by splitting the model into two parts: one trained locally by each client and one trained jointly by all clients. The novelty lies in splitting the model to have more encoder layers trained locally and fewer globally. To optimize communication, only a subset of clients participates in each global update, using the FedAVG algorithm to train the global layers.

*Yu, Muñoz & Jannesari (2023)* introduced a method to select a subset of parameters from the main model to be used in FL using saliency maps. The L1-norm is used to rank

the original model's weights, and only a small percentage of these weights, determined by thresholding the L1 score, are sent to the server. The server performs FedAVG on the received features and sends only these features back to the clients. The authors demonstrate that this approach accelerates training of FLs in FL by nearly halving the computational time, while maintaining and even improving performance. This suggests that performing FedAVG on a subset of parameters can act as regularization, similar to stochastic weight averaging (*Guo, Jin & Liu, 2023*).

### Test-time adaptation for training

The methods in this class only optimize the input to the FMs, by collaboratively adjusting it using FL (*Yang, Wang & Wang, 2023*).

*Guo et al. (2023)* introduced a method for collaboratively training prompts instead of the entire model. Their approach assumes that each client pre-trains or uses an off-the-shelf FM on their local data, and uses FL to optimize continuous prompts, which are learnable vectors directly integrated into the model's embedding space. The method introduces a new set of learnable input parameters representing the continuous prompt and optimizes only these parameters in FL using FedAVG. While this significantly reduces communication costs, it presents additional challenges, as it assumes clients can pre-train models independently on their data. Additionally, adjustments are needed, as the number of clients involved in optimizing the prompts directly affects performance. *Su et al. (2022)* proposed a similar method where prompts are trained collaboratively, but the contribution of each client's prompt is modulated by a learnable parameter called a key. This technique mirrors the aggregation methods used in training entire models, but is specifically applied to prompts. Using the learnable weight (key) enables a more personalized aggregation process, where clients with similar domains, but different distributions, can customize their contribution to the final prompt. *Zhao et al. (2023)* extended the experiments with continuous prompts to evaluate whether the exchange of prompts can compromise the privacy of FL, and found that introducing prompt tuning in FL does not inherently breach privacy.

## Customize

Customizing existing FMs using FL is one of the most attractive use cases, as it is more efficient to train a FM in a centralized manner or to use an off-the-shelf pre-trained FM and then customize it using FL. This approach aligns well with the ML development life-cycle, where starting from pre-trained models has become a standard practice for most tasks. Therefore, the algorithms in this class are also the most numerous. To initially classify them, we first consider the type of method used for customization: (i) *fine-tuning*—where parts or the entire model are fine-tuned in FL using supervised or semi-supervised learning, (ii) *contraction*—where the size of the model is reduced using various techniques, and (iii) *hybrid* methods—combining fine-tuning and contraction. On the server side, most algorithms discussed in this section use FedAVG or its variants for parameter aggregation. Within these classes, several sub-categories can be identified, which are detailed in the following sections.

## Fine-tuning

Fine-tuning methods involve continuing to train a FM in FL using supervised or semi-supervised learning. Given that fine-tuning the entire model is both computationally and communication-intensive, most methods in this class focus on fine-tuning only a selection of existing or new parameters. To further distinguish between these methods, we classify them into two categories: methods that select only parts of the model for fine-tuning (*selective*) and methods that add extra parameters for fine-tuning (*additive*), both of which are presented in the following paragraphs.

*Selective.* Selective methods fine-tune only a subset of the existing parameters of the FM using FL. Unlike pre-training with selective methods, the methods in this class use supervised or semi-supervised learning and primarily focus on fine-tuning the bias parameters of a model in FL, while keeping the rest of the parameters frozen. *Sun et al. (2022)* are among the firsts to evaluate the performance of fine-tuning only the bias parameters (also known as bias tuning) and compare it to adding extra adapters under various settings, including distinct client stability, data distributions, and differential privacy settings. During training, each client performs a forward pass through the model, sends the bias parameters to the server, which aggregates them using FedAVG and returns the outcome to all clients. The authors found that bias tuning consistently outperforms additive fine-tuning, across distinct use cases involving both language and vision. This performance gap is maintained across various settings, such as when differential privacy is used for training, or when all clients operate in low-data regimes. *Chen et al. (2022a)* ran a similar study, with similar conclusions, using different types of adapters.

*Tsouvalas, Asano & Saeed (2024)* presented a novel technique that selects a subset of network parameters based on the L1 norm of the weight matrices at each layer. The algorithm ranks these matrices using the L1 norm and aggregates only the top $n$ matrices, with $n$ being determined by the specific use case. This approach significantly reduces the number of parameters involved in the aggregation process. The authors demonstrated a reduction of more than 60% in the number of parameters for large NLP architectures such as BERT-Large. However, they obserdve a less pronounced decrease for smaller models, indicating that the effectiveness of this technique may vary depending on the model size and complexity.

*Additive.* Additive methods introduce extra parameters that are fine-tuned while keeping the FM frozen. These methods focus on adding extra parameters in the final layers, typically called adapters, which project the final embeddings of the FM and adapt them to new tasks, or on using parameter-efficient fine tuning (PEFT) (particularly LoRA—described in Appendix) to add parameters at each layer. When using LoRA, a common question arises: should the added LoRA ranks be homogeneous or heterogeneous, given that some clients may have better computational resources or larger datasets. Additionally, within LoRA, there is the possibility of adding asymmetric parameters between the low-rank matrices. An illustration of these methods is provided in Fig. 5, where Fig. 5A shows the addition of extra

adapters, Fig. 5B illustrates the use of LoRA with either homogeneous or heterogeneous (gray) parameters, and Fig. 5C illustrates the use of asymmetric LoRA parameters.

*Houlsby et al. (2019)* were the first to analyze which parameters of a FM should be fine-tuned for better precision by comparing fine-tuning directly the last layers of a FM with adding one extra layer to fine-tune (Fig. 5A). The authors found that adding extra adapters not only improves the performance, but also leads to more stable training. *Chen et al. (2022a)* tested this idea in FL settings and found that using an adapter does improve the performance but only in some cases. In other cases, selecting and fine-tuning only the bias parameters of a transformer leads to better results (discussed in the selective paragraph from above).

*Legate et al. (2024)* extended this idea and proposed to first fine-tune only the extra adapter, followed by some steps in which the complete model is fine-tuned using FedAVG. This second step is shown to further improve performance, however, it increases the communication costs significantly.

When using LoRA in FL, a key consideration is whether to make the added low-rank parameters homogeneous (identical) across clients or heterogeneous. Several studies use homogeneous LoRA, as discussed in Appendix, for distinct tasks. *Nguyen, Munoz & Jannesari (2024)* used it to fine-tune a vision-language model where the encoders for either the vision or language components are extended using LoRA. *Jiang et al. (2024)* used LoRA to fine-tune large language models (LLMs), while *Zhang et al. (2024a)* used it for instruction-tuning LLMs in FL. *Yi et al. (2023)* applied it in scenarios where clients own heterogeneous FMs, but use homogeneous parameters for fine-tuning, by manually selecting the layers for parameter addition. In all these cases, the aggregation of global (homogeneous) LoRA parameters is performed using FedAVG, with the mentioned studies showing consistent performance improvements across different benchmarks and modalities.

However, using homogeneous LoRA ranks presents a trade-off between overfitting and slow convergence, especially for clients with more heterogeneous datasets or models (*e.g.*, skewed distributions) or when personalization is required. In such cases, the definition and aggregation of local parameters for LoRA can be made more dynamic by introducing heterogeneous parameters. For example, *Guo et al. (2024)* fine-tuned a multi-language LLM by introducing heterogeneous local adapters for each language family (*e.g.*, Germanic or Italic), and aggregating only the parameters relevant to a specific language family globally. This approach helped mitigate inter-language bias and offered flexibility for more challenging languages, at the expense of increased communication costs.

Additionally, *Cho et al. (2023)* proposed a method to assign and aggregate heterogeneous ranks to all clients based on their system capabilities (*e.g.*, distributing higher ranks to more capable clients and *vice versa*). The challenge then becomes aggregating the heterogeneous parameters into global LoRA parameters. To achieve this, the server pads all ranks to the same size and applies weighting based on the norm of the singular value vectors for the local parameters. The padded parameters are then aggregated using weighted FedAVG. This not only improves communication efficiency, as smaller parameter exchanges occur for some clients, but also acts as a form of regularization that enhances final performance

compared to homogeneous LoRA. *Byun & Lee (2024)* further investigated the padding-based aggregation of heterogeneous ranks and discovered that it introduces instabilities during training. To mitigate these instabilities, the authors used replication-based padding instead of zero-padding, which led to more robust and efficient fine-tuning.

More complex approaches, which combine multiple sets of homogeneous and heterogeneous adapters have also been explored. *Yang et al. (2024b)* proposed using two sets of LoRA parameters: a homogeneous set that is aggregated and updated globally, and a heterogeneous that is kept local. This technique can address skewed distributions between the client datasets or facilitate personalization. During each training step, the forward pass uses the frozen FM, the global adapter, and the local adapter. To balance the contribution of the global and local adapters, a weighting mechanism can be additionally employed. The global adapter is aggregated by the server using FedAVG, while the local adapter is kept individual for each system. This technique provides further adaptability without compromising the global performance.

*Ping et al. (2024)* explored this concept in a multi-task setting by defining multiple (heterogeneous) LoRA parameters for each task. During each training step, only the parameters specific to a task are aggregated using FedAVG. In scenarios where task labels are not explicitly defined, the authors introduced a mechanism that employs k-means clustering to group and aggregate the parameters. This approach automatically clusters similar adapters from the clients, which are assumed to contribute to the same tasks, and aggregates them using FedAVG. The method demonstrated improved robustness in multi-task settings by mitigating cross-task drift. However, it incurs high computational costs.

*Tian et al. (2024)* introduced additional heterogeneity by modifying one of the low-rank matrices from LoRA ($A$, $B$) to add multiple asymetric matrices for $B$ (illustrated in Fig. 5C). This approach allows specialization for particular tasks in multi-task settings, similar to mixture of experts (MOE) (*Cai et al., 2025*). During each training step, the matrix $A$ is aggregated globally using FedAVG, while each part of matrix $B$ is aggregated only with its corresponding part from other clients, also using FedAVG. The method demonstrated consistent performance improvements even when the combined sizes of the asymmetric matrices are equal to the initial size of $B$, providing evidence that specializing these matrices for particular tasks can improve overall performance without increasing the parameter count.

In addition to parameter homogeneity, *Babakniya et al. (2023)* found that the initialization of the LoRA parameters can introduce instabilities and affect performance, as a single initialization may not suit all client data distributions. To mitigate these issues, the authors proposed to initially fine-tune the entire model for a few steps, followed by fine-tuning only the LoRA parameters. This approach provides initialization and momentum from the entire data distribution from all clients, which represents a better initialization. However, fine-tuning the entire model comes with high computational costs and may not always be feasible. *Wu et al. (2024b)* extended this procedure by alternating between full fine-tuning and fine-tuning only the LoRA parameters. This method aims to

balance global and local knowledge exchange, demonstrating performance improvements but at an even higher cost.

*Yang et al. (2024a)* further advanced this idea by introducing a process similar to simulated annealing. They proposed initially fine-tuning the entire model using FedAVG while applying the L2 norm to all parameters to mitigate potential client drifts. In the second stage, fine-tuning is limited to the LoRA parameters, but the ranks are adaptively decreased during training. This stage begins with more parameters in the early epochs, gradually reducing the number of parameters (ranks) as training progresses, similar to a learning rate scheduler. However, the method adds overhead for adjusting the ranks and only improves communication efficiency in the final stages when the number of parameters decreases.

Instead of fine-tuning the entire FM, *Yan et al. (2024)* suggested initializing the LoRA ranks using the results of singular value decomposition on the pre-trained weight matrices. This approach introduces only a minor initialization step which helps stabilize the entire training process.

*Zhang et al. (2023b)* conducted an empirical study on additive fine-tuning in FL and found that, compared to fine-tuning the entire model, additive fine-tuning offers better defense against certain threat models, such as data reconstruction attacks. Additionally, they discovered that adaptive fine-tuning generally performs on par with complete fine-tuning while significantly reducing communication costs, by factors ranging from 12 to 190 times. However, additive fine-tuning is more sensitive to data heterogeneity.

## Contraction

Contraction methods aim to decrease the size of the models to optimize communication and computation efficiency. The methods in this class are further classified in methods that transfer the knowledge from one model to another (distillation) and methods that aim compress the FM (through compression, pruning, or quantization).

*Distillation.* As discussed in the Appendix, KD in FL involves transferring knowledge from local models to a global, in this case, FM. This process is equivalent to training local models on local datasets and then distilling this knowledge into the FM using a shared dataset (*Wu et al., 2022*). Additionally, it is possible to distill the local datasets by training synthetic data generators. This synthetic data can either be shared directly with the global model or used to distill the knowledge from the local models to the global model. Figure 6A illustrates training local models on private data and distilling the knowledge on public data and Fig. 6B illustrates the scenario where a synthetic data generator is trained using locally trained models. While the intersection of KD and FL has been explored, we focus on presenting papers that utilize use FMs or aim to customize pre-trained models. For a more comprehensive overview of KD and FL, we refer readers to the work of *Li et al. (2024a)*.

*Gong et al. (2021)* are among the first to customize a FM using ensemble KD (Fig. 6A). In the first stage of their approach, each client trains a local model on their private datasets. In the second stage, the local datasets are disconnected, and a central public dataset is used for distillation. At each training step, the server distributes data to each node, which
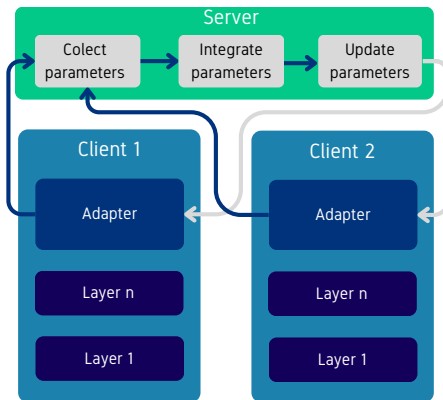

**(a)** Additive fine-tuning with adapters in the final stage, where only the final blocks are added and adjusted in federated learning. These blocks can be one or multiple layers.

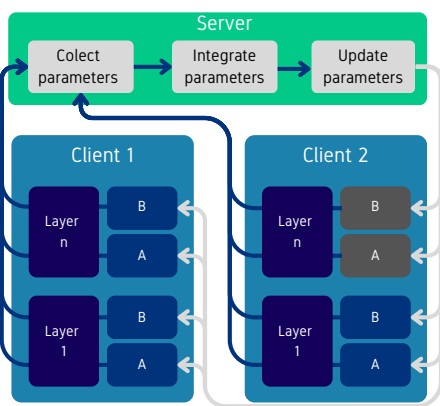

**(b)** Additive fine-tuning using homogeneous LoRA adapters, where at each layer a set of homogeneous LoRA parameters are defined and adjusted in federated learning.

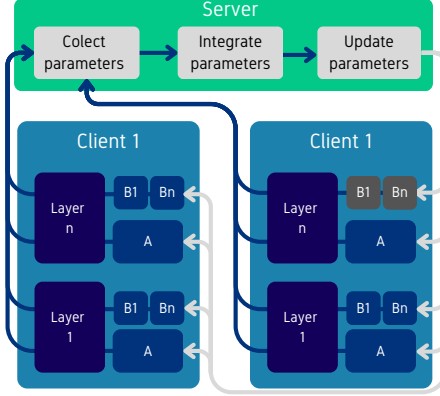

**(c)** Additive fine-tuning using asymmetric LoRA, where at each layer a set of heterogeneous LoRA parameters are defined and adjusted in federated learning. These parameters can be heterogeneous depending on the layers or on the clients.

**Figure 5** Overview of additive fine-tuning methods using federated learning, where for each type of methods the figures show which parameters are communicated from the clients toss the server.

runs inference on the local model and sends the results back to the server. The server then aggregates these results using FedAVG and uses them to distill the knowledge to the FM using the KL-divergence as described in the "Knowledge distillation" section. Additionally, beyond performing KD only with the final outputs, the authors introduce a mechanism to also distill the attention from transformer layers, which improves the overall effectiveness of the knowledge transfer by transferring intermediate features as well. While the method demonstrates performance gains, especially when compared to training the model using fine-tuning with FedAVG, it introduces a dependency on the availability of a dataset, which should be sufficiently large to enable KD between the local nodes and the global FM. While the method demonstrates performance gains, particularly when compared to fine-tuning

with FedAVG, it introduces a dependency on the availability of a sufficiently large public dataset, that can facilitate KD between the local nodes and the global FM.

*Wu et al. (2023)* assume the existence of a FM for each client node, assuming the clients can pre-train their own models or use off-the-shelf FMs. In this setting, each client keeps the FM private and trains a small-scale proxy model used to exchange the knowledge. The proxy models are aggregated by the server using FedAVG. *Wu et al. (2023)* assume the existence of a FM for each client node, either by allowing clients to pre-train their own models or by using off-the-shelf FMs. In this setting, each client keeps the FM private and trains a small-scale proxy model used to exchange knowledge. At each training step, KD is used between the FM and the proxy. The proxy models from each client are then aggregated by the server using FedAVG. This approach ensures that the knowledge from the private FMs is shared and integrated into the global model without sharing the models. However, it assumes the existence of FMs at each client, which may not always be realistic.

*Zhang et al. (2022b)* introduce a method where the server trains a synthetic data generator used for KD, as illustrated in Fig. 6B. In this framework, each client trains their own local model and shares it with the server. The server then trains a synthetic data generator using these local models, aiming to match the semantics of the outputs from the local models with synthetic data. Subsequently, the generator is used to produce synthetic data, perform inference on the local models, and distill this knowledge to the global model using KD (as described above). This method ensures that the knowledge from the local models is transferred to the global model without the need to share the original data. However, it introduces additional failure dependencies on the quality of the synthetic data generated, which can impact performance in different scenarios (*Zhang et al., 2022a*) present a similar framework where the server trains a data generator. However, in their approach, the generator is also trained using KD, instead of using a semantic loss.

For language-specific tasks, *Deng et al. (2023)* do not train a new generator but instead use the FM directly to generate new data. In this framework, each client trains a small model on their local, private data. The clients then transfer their models to the server, which uses them to calibrate the generation process and synthesize a new dataset. This dataset is subsequently used for KD, as described above.

*Compression, pruning, and quantization.* While techniques such as compression, pruning, and quantization have been extensively studied for reducing model size and implicitly lowering communication costs in both FL and FMs, their intersection has received relatively little attention in the literature. Compression involves either compressing the weights of local models (*e.g.*, using singular-value decomposition (*Yang, 2021*)) before transmission to the server, or compressing the gradients for back-propagation (*e.g.*, using stochastic sign-based methods (*Tang, Wang & Chang, 2024*)) and aggregating them on the server. Pruning involves iteratively removing model parameters with small values during collaborative training (*Jiang et al., 2022*), while quantization reduces the number of bits used to represent the weight matrices (*Chen et al., 2022b*).

As these concerns can be considered together with other customization techniques such as fine-tuning techniques, most of the works for FMs using compression, pruning,

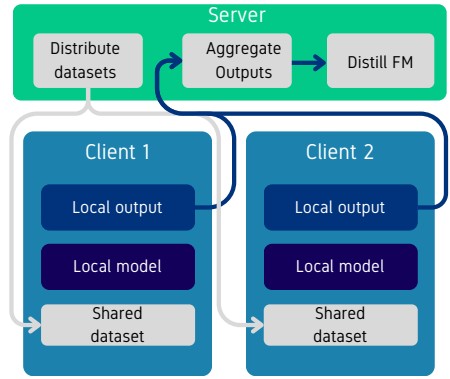
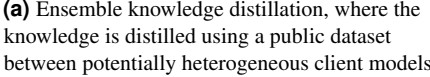

(a) Ensemble knowledge distillation, where the knowledge is distilled using a public dataset between potentially heterogeneous client models.

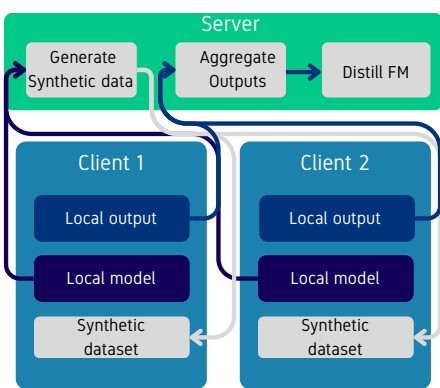

(b) Generative knowledge distillation, where clients collaborate to also generate a synthetic dataset used for distillation.

**Figure 6** Overview of knowledge distillation methods using federated learning.

or quantization will be discussed in the *hybrid* section. The only study on FMs without additional techniques is by *Yang et al. (2023)*, who employ quantization to compress the weight matrices (excluding other parameters like biases) in a large FM from 32 to 16 bits. To mitigate the effects of quantization, the authors propose applying a linear (learnable) transformation on the server side when integrating the aggregated parameters. Additionally, they suggest alternating and quantizing only a subset of parameters for each client, varying the selection across clients to ensure the server receives precise updates from clients that did not quantize those parameters. Using these techniques, they achieve communication cost reductions of over 60% in some cases, assuming that clients can perform inference on the FM locally and the server can run back-propagation for the quantized parameters.

*Hybrid.* Hybrid methods use both fine-tuning and contraction or multiple contraction methods methods to further improve communication costs.

*Chen et al. (2022b)* use selective fine-tuning with quantization to reduce communication costs. Instead of selecting individual weights, the authors propose partitioning the FM into blocks containing multiple layers and selecting only some of these blocks for fine-tuning in FL. At each training step, each block is assigned an importance score using the average L2 norm between the block values from the last two epochs. The score is used to select only the top $n$ blocks, where $n$ is determined based on the task. To further reduce communication costs, the selected blocks are quantized using stochastic quantization (*Dong et al., 2019*) before being sent to and aggregated on the server (using FedAVG). This method demonstrates consistent performance improvements and reduced communication costs compared to other selective fine-tuning approaches.

*Chen et al. (2024)* introduce a hybrid approach that combines adapter-based fine-tuning with knowledge distillation. Similar to the dual adapter methods discussed in the additive fine-tuning section, each client maintains a global adapter and a local client-specific adapter. During each training epoch, clients share the global adapter, which is aggregated

by the server and then redistributed to the clients. This global adapter is used to distill its knowledge into the local adapter using the client's local dataset. The proposed method demonstrates consistent performance improvements over traditional adaptive fine-tuning or distillation techniques. However, it introduces additional computational constraints due to the local distillation process. *Kuo et al. (2024)* prune the LoRA parameters before uploading and aggregating them on the server by applying a magnitude-based threshold. Using the shared parameters, the server generates a global sparsification mask, averages the parameters using FedAVG, and then distributes the parameters back to the clients. The clients update only the parameters corresponding to the sparsification mask, retaining all other parameters as they were before sparsification, enabling dense training. This method demonstrate an order of magnitude improvement in communication costs while maintaining competitive performance with full LoRA fine-tuning. *Yadav et al. (2023)* improve this method by incorporating quantization of the sparsified parameters, resulting in further size reduction.

*Wu et al. (2024a)* propose a method that combines compression with distillation and adaptive fine-tuning. Initially, the server compresses the FM into a more compact model by extracting a sub-model using layer dropout. This sub-model acts as a student model to distill information from the FM using a public dataset. The compressed model is then used with LoRA and trained within in FL, as described in the adaptive fine-tuning section.

## Deploy

Deploying models through FL involves performing inference at each client using their local models, rather than relying on a final global model. This approach can be viewed as delivering an inference service leveraging the FL infrastructure (*Han et al., 2024*).

The only study that addresses the deployment of FMs using FL is the work by *Liu et al. (2022)*. The authors proposed projecting multi-scale embeddings from intermediate layers and accumulating these projections across multiple clients to perform predictions. For example, in a classification task using transformer architectures, the class token responsible for classification is extracted from intermediate layers. The choice of layer is determined by the available resources at each client. Clients with fewer resources can use the token from earlier layers, while those with more resources can use the token from later layers. The final prediction is an average of the client predictions.

## FMS AND FL IN HEALTHCARE

Healthcare applications are among the most compelling use cases for FL (*Pfitzner, Steckhan & Arnrich, 2021*), and represent a significant area where FMs can make a substantial impact (*Moor et al., 2023*). However, despite the potential, relatively few articles have explored the use of FMs and FL in healthcare applications.

*Manoel et al. (2023)* fine-tuned a multilingual FM for medical transcript analysis. At each epoch, the clients fine-tune the entire model using their own datasets for a specific number of batches and upload the entire model to the server. The only optimization introduced is to train the model locally for a larger number of batches, to decrease the communication costs (similar to *Sani et al. (2024)*). *Wang et al. (2024)* introduced a new

dataset with eight distinct medical tasks, including classification, anomaly detection, and generative tasks, designed for fine-tuning FMs using FL. The authors conduct a comparative benchmark using this dataset. In their setup, each client trains a local model on their private data and uploads it to a server, which employs FedAVG to aggregate these models into a global model. The global model is then used to incorporate local knowledge into a FM by concatenating the global model's output with the input of an FM. Using a public dataset, the FM is fine-tuned together with the local knowledge. Using this method, the authors demonstrate performance improvements across all tasks in the dataset. *Liu, Luo & Zhu (2024)* used FL to develop a foundational segmentation model based on the segment anything method (*Kirillov et al., 2023*). The authors perform additive fine-tuning, where each client fine-tunes additional lightweight adapters for the segment anything model using their local data. The adapters are then aggregated on the server using FedAVG. Empirical experiments show that the performance achieved using FL is comparable to that of centralized training, while significantly reducing the resources required for training.

## PRACTICAL PERSPECTIVES

From a practical standpoint, choosing the right algorithms to experiment with can be challenging. This is because many algorithms are interconnected, and their trade-offs beyond performance are not always evident. For example, it can be difficult to differentiate between various additive fine-tuning methods and decide which one to prototype first or trace its incremental development. Furthermore, comparing the algorithms' performance is not always relevant, as they are tested on distinct datasets and modalities.

To assist with this issue, we present an initial classification of the methods discussed in this paper, using three criteria: complexity, efficiency, and scalability. Each criterion is defined on a scale ranging from 1 to 3, denoted as * to *** in Table 3.

First, we evaluate the complexity of the method, which is estimated based on the effort required to implement it. For incremental methods, we consider the base method to have the lowest complexity, with any additional features or modifications increasing its complexity. The extent of this increase depends on the nature and complexity of the additions. After classifying the methods based on their inherent complexity, we compared and scaled the base methods accordingly. For example, complexity * indicates minimal implementation effort such as adding a classifier head, ** indicates moderate effort such as using more complex adapters such as LoRA, and *** indicates a large effort, such as having client-specific, hybrid, adapters.

Second, we assess the efficiency of the method, which is measured by the amount of information transmitted from the clients to the server. Here, we follow a similar approach to reduce iterative methods to their base method and first compare the base methods. For example, * indicates high communication overhead, such as updating and communicating all parameters of a model, ** indicates partial parameter updates such as updating and communicating adapter layers, and *** indicates highly efficiency methods such as hybrid methods that communicate a small number of parameters by quantization or compression.

**Table 3** **Classification of the methods discussed based on their complexity, efficiency, and scalability, and following the taxonomy from Fig. 2.** For complexity, * denotes low implementation effort (*e.g.*, adding a classifier head), ** denotes moderate effort (*e.g.*, using adapters like LoRA), and *** denotes high effort (*e.g.*, client-specific hybrid adapters). For efficiency, * indicates high communication overhead (*e.g.*, full model updates), ** indicates moderate overhead (*e.g.*, adapter updates), and *** indicates highly efficient methods (*e.g.*, quantized or compressed updates). For scalability, * denotes low scalability, ** moderate scalability, and *** high scalability.

| Technique | | Complexity * | Complexity ** | Complexity *** | Efficiency * | Efficiency ** | Efficiency *** | Scalability * | Scalability ** | Scalability *** |
|---|---|---|---|---|---|---|---|---|---|---|
| Pretrain | Entire | Garcia Bernal et al. (2021), Sani et al. (2024), Rehman et al. (2023) | Zhuang et al. (2021), Li, He & Song (2021) | Makhija, Ho & Ghosh (2022), Recasens et al. (2023) | Garcia Bernal et al. (2021), Li, He & Song (2021), Makhija, Ho & Ghosh (2022), Rehman et al. (2023) | Sani et al. (2024), Recasens et al. (2023) | Zhuang et al. (2021) | Garcia Bernal et al. (2021), Li, He & Song (2021), Makhija, Ho & Ghosh (2022), Rehman et al. (2023), Recasens et al. (2023) | Sani et al. (2024), Zhuang et al. (2021) | |
| | Partial | Tan et al. (2022), Chen, Su & Li (2024), Kim, Yoo & Kang (2023), Lit et al. (2022), Yu, Muñoz & Jannesari (2023a) | Lu et al. (2024) | | Lu et al. (2024) | Tan et al. (2022), Kim, Yoo & Kang (2023), Lit et al. (2022) | Chen, Su & Li (2024), Yu, Muñoz & Jannesari (2023a) | Lu et al. (2024) | Tan et al. (2022), Chen, Su & Li (2024), Kim, Yoo & Kang (2023), Lit et al. (2022), Yu, Muñoz & Jannesari (2023a) | |
| | Test-time ad. | | Guo et al. (2023), Su et al. (2022), Zhao et al. (2023) | | | Guo et al. (2023), Su et al. (2022), Zhao et al. (2023) | | | Guo et al. (2023), Su et al. (2022), Zhao et al. (2023) | |
| Customize | Fine-tune | Sun et al. (2022), Chen et al. (2022a), Tsouvalas, Asano & Saeed (2024), Houlsby et al. (2019) | Legate et al. (2024), Nguyen, Munoz & Jannesari (2024), Jiang et al. (2024), Zhang et al. (2024a), Yi et al. (2023), Yang et al. (2024a), Yan et al. (2024), Zhang et al. (2023b) | Guo et al. (2024), Cho et al. (2023), Byun & Lee (2024), Yang et al. (2024b), Ping et al. (2024), Tian et al. (2024), Babakniya et al. (2023), Wu et al. (2024b) | Ping et al. (2024), Babakniya et al. (2023), Wu et al. (2024b), Yang et al. (2024a) | Sun et al. (2022), Chen et al. (2022a), Legate et al. (2024), Nguyen, Munoz & Jannesari (2024), Jiang et al. (2024), Zhang et al. (2024a), Yi et al. (2023), Guo et al. (2024), Cho et al. (2023), Byun & Lee (2024), Yang et al. (2024b), Tian et al. (2024), Yan et al. (2024), Zhang et al. (2023b) | Tsouvalas, Asano & Saeed (2024), Houlsby et al. (2019) | Legate et al. (2024), Ping et al. (2024), Tian et al. (2024), Babakniya et al. (2023), Wu et al. (2024b), Yang et al. (2024a) | Sun et al. (2022), Chen et al. (2022a), Nguyen, Munoz & Jannesari (2024), Jiang et al. (2024), Zhang et al. (2024a), Yi et al. (2023), Guo et al. (2024), Byun & Lee (2024), Yang et al. (2024b), Yan et al. (2024), Zhang et al. (2023b) | Tsouvalas, Asano & Saeed (2024), Houlsby et al. (2019), Cho et al. (2023) |
| | Contraction | Yang et al. (2023) | Gong et al. (2021) | Wu et al. (2023), Zhang et al. (2022b), Zhang et al. (2022a), Deng et al. (2023) | Wu et al. (2023), Zhang et al. (2022b), Zhang et al. (2022a) | Gong et al. (2021), Deng et al. (2023) | Yang et al. (2023) | Gong et al. (2021), Wu et al. (2023), Zhang et al. (2022a) | Zhang et al. (2022b), Deng et al. (2023) | Yang et al. (2023) |
| | Hybrid | Chen et al. (2022b) | Chen et al. (2024), Kuo et al. (2024), Yadav et al. (2023) | Wu et al. (2024a) | | Chen et al. (2024), Wu et al. (2024a) | Chen et al. (2022b), Kuo et al. (2024), Yadav et al. (2023) | Wu et al. (2024a) | Chen et al. (2024) | Chen et al. (2022b), Kuo et al. (2024), Yadav et al. (2023) |
| Deploy | | Liu et al. (2022) | | | Liu et al. (2022) | | | Liu et al. (2022) | | |

Third, we consider the scalability of the method, which is estimated based on the resources required for both the client and the server. For example, if a method requires clients to run a FM on their premises, it will have lower scalability compared to methods that require inference on smaller models. For example, * indicates heavy computation on client or server, such as pre-training a large model, ** indicates moderate compute such as performing inference and optimization on a subset of parameters, and *** indicates low-resource needs such as test time adaptation. This classification system offers initial guidance in choosing methods to experiment with first, based on project constraints, and later on progressing to more complex methods. To assign the scores, we used a process similar to validating the taxonomy, where two experts assigned individual scores, and potential disagreements were resolved through discussion and consensus.

We observe that, for pre-training FMs with FL, the methods used for pre-training the entire model exhibit the lowest scalability but also low complexity. This is because basic algorithms can be employed, but the resources required to run training at each client and aggregate the results are substantial. Similarly, methods that assume clients can independently train FMs, such as those described in *Makhija, Ho & Ghosh (2022)*, have higher complexity and low scalability. In general, pre-training FMs in FL demands significant resources, as either the clients or the server must perform forward passes on the model, which are generally computationally intensive. Therefore, from a practical standpoint, it is best to begin with partial model training, particularly using selective methods that can be both efficient and scalable. Test-time adaptation methods are suitable only for certain types of models, such as large language models or text-based models.

For customizing FMs in FL, we observe that methods using adapter fine-tuning with small adapters, such as adding a final layer, are the least complex and the most scalable and efficient. These methods also provide good performance, making them a recommended start point.

More complex additive methods, such as those using LoRA, require forward passes of the entire model at the clients while aggregating only the additive parameters. This introduces additional complexity in managing the extra parameters. Similarly, more advanced LoRA methods that employ heterogeneous or dual adapters further increase complexity, although some, like those described in *Cho et al. (2023)*, may offer scalability improvements.

From the contraction methods, compression and quantization algorithms are the fastest to prototype, as they have lower complexity and high efficiency and scalability. Moreover, these algorithms can be used in conjunction with any other types of algorithms, making them highly versatile. The KD algorithms are relatively complex. They assume either the existence of an independent dataset or the ability to train independent FMs or data generators for each client. These constraints make KD algorithms less practical for many use cases, as they require additional resources and infrastructure.

## STUDY LIMITATIONS

While this literature survey aims to provide objective insights into FMs and FL, several limitations must be acknowledged that may influence the generalizability of the findings:

- Selection bias: The inclusion of studies was guided by predetermined search terms, databases, and criteria for inclusion and exclusion. As a result, there is a potential for selection bias, which could have led to the omission of relevant studies that did not meet these criteria. We tried to avoid this bias by including major information sources such as Google Scholar or ScienceDirect.
- Timeframe constraints: The scope of this survey was limited to studies published within a defined period, potentially excluding early works or very recent advancements in the field. This could impact the completeness of the review, especially in a rapidly evolving research area such as machine learning. Nevertheless, given that the uptake of FMs significantly increased after 2022, this article is primarily subject to the risk of missing the most recent developments.
- Exclusion of grey literature: The survey primarily focused on peer-reviewed or academical articles, excluding non-peer-reviewed sources such as theses, technical reports, or more general blog articles. This exclusion of grey literature may have resulted in missing insights and contributions from practitioners, especially regarding the discussion providing practical perspectives.
- Subjectivity in interpretation: The process of reviewing and interpreting the literature involves a degree of subjectivity. Despite efforts to adhere to objective criteria, such as asking opinions from othre independent researchers, researcher bias may have influenced decisions regarding the inclusion of studies, the interpretation of findings, or the ranking and comparison of methods.
- Publication bias: This survey may be subject to publication bias, where studies with significant or positive results are more likely to be published, thus potentially over-representing certain outcomes while underrepresenting null or negative results.

## DISCUSSION

In this section, we address and provide responses to the initial research questions outlined in Table 2. An graphical representation of the findings is summarized in Table 4.

To address RQ1, we identified approximately 48 distinct methods that combine FMs with FL at various phases of the development life-cycle. The majority of these methods fall under the category of FM customization through fine-tuning. Moreover, most of the presented methods use relatively straightforward FL and machine learning primitives. They predominantly rely on FedAVG for the aggregation mechanism and on simple adjustments to established algorithms such as using contrastive learning for pre-training, LoRA for fine-tuning, or KD. This presents an opportunity for future work, as exploring more adaptive aggregators (*e.g.*, FedAdam) or innovative pre-training techniques could yield more efficient and effective integration of FM with FL.

**Table 4** Summarized answers to the research questions that guided this study, as outlined in Table 2.

| ID | Answer |
| --- | --- |
| RQ1 | Pre-training entire models or a subset of model parameters |
| | Fine-tuning using new adapters in the final stage or for all layers |
| | Knowledge distillation using a shared or a synthetic dataset |
| RQ2 | Communication efficiency vs. model complexity |
| | Personalization vs. generalization |
| | Integration of privacy enhancing technologies |
| RQ3 | Additive fine-tuning with adapters in the final stage or LoRA |
| | Hybrid approaches, combining compression, quantization, and additive fine-tuning |
| RQ4 | Entire model fine-tuning |
| | Additive fine-tuning with adapters in the final stage |
| RQ5 | Development and use of advanced aggregation mechanisms |
| | Integration of privacy enhancing technologies |
| | Applications and trade-off assessment in practical use-cases |
| | Standardized benchmarking |

To address RQ2, the literature highlights and amplifies the trade-offs inherent in FL when using FMs. One of the primary concerns is the trade-off between communication efficiency and model complexity. Communicating large model updates in FL can substantially increase training time. To alleviate this trade-off, efficient customization techniques have been explored. These include selective or additive model updates, where only a fraction of the FMs parameters are adapted, as well as hybrid techniques such as model compression or pruning. Another FL trade-off exists between personalization and generalization, where some clients may require models tailored to their specific data, while others may prioritize generalization to diverse datasets. Fine-tuning FMs can balance these competing needs, as robustness is, in principle, assured when using pre-trained models. We also observed that only a small subset of the reviewed publications investigates additional privacy-preserving techniques, such as differential privacy or secure multiparty computation. Incorporating these methods could further enhance privacy at the cost of performance, introducing another important trade-off to consider, and an opportunity for future work.

To address RQ3, as discussed in the Practical Perspectives section and Table 3, certain methods offer low complexity and high scalability or efficiency, such as the methods by *Sun et al. (2022)* or *Chen et al. (2022b)*. Overall, the most compelling methods use a hybrid approach, addressing distinct trade-offs through multiple techniques. These include reducing the number of parameters used and compressing them during communication. However, very few methods excel in achieving low complexity, high efficiency, and high scalability simultaneously, leaving ample room for future work.

To explore the application of FL and FMs, we focused on the healthcare domain as a use case. Our goal was to investigate whether these methods have been adopted and studied in real-world scenarios. Healthcare was selected as it is widely considered an ideal use

case for FL, given that hospitals often face data privacy challenges and can benefit from collaborative model training without sharing sensitive data. However, to address RQ4, our literature review revealed only a limited number of publications in this area. This scarcity of research opens up new avenues for future work, presenting opportunities to further investigate and apply these methods in practical settings, as well as investigate and discover novel potential trade-offs.

Lastly, to address RQ5, and as previously indicated, our literature analysis revealed several promising research directions. First, studying and developing novel aggregation mechanisms has the potential to improve the performance and efficiency of FL and FMs, where advanced techniques such as adaptive aggregation, hierarchical aggregation, or personalized aggregation can improve the convergence speed, performance and robustness. Second, further in-depth studies are essential to understand the trade-offs between model size and efficiency, as well as privacy and performance, when integrating FMs and FL. Such investigations can identify potential privacy limitations and their impact on overall system efficiency. For instance, while techniques like LoRA reduce communication costs, they still require a forward pass over the entire model, including additional adapters, during each iteration. This can limit clients to using smaller FMs due to computational constraints. Similarly, the effectiveness of differential privacy techniques may be diminished when applied to pre-trained models, as these models might retain more sensitive information from pre-training. Third, there is a need for more practical use-cases to evaluate the generalizability and robustness of available methods. By exploring a wider range of real-world applications, researchers can determine whether these methods perform consistently across different domains and scenarios, or whether novel trade-offs are identified. Last, this can lead to more comprehensive benchmarks and integrated datasets for evaluating these algorithms, which are not currently evaluated on similar data.

Furthermore, several challenges remain unsolved or underexplored. Achieving a balance between personalization and generalization continues to be a significant challenge. While many methods enhance performance on client-specific data, few provide mechanisms to ensure robustness and generalizability across heterogeneous participants or tasks.

Although privacy is a core motivation for adopting FL, most reviewed methods lack integrated privacy guarantees such as differential privacy or secure aggregation through encription—an omission that is especially concerning in sensitive domains like healthcare. Additionally, existing methods often fail to reconcile the trade-offs between scalability, communication efficiency, and implementation complexiry. Techniques tend to prioritize one of these aspects at the cost of the others, thereby limiting their practical deployment.

While the medical domain is frequently cited as a key application area, real-world implementations of FL–FM systems in healthcare settings remain scarce, with few methods evaluated on clinical datasets or tested in real deployment scenarios.

## CONCLUSIONS

We conducted a comprehensive literature review on FMs using FL, covering all stages of the development life cycle, including training, customization, and deployment of models. Over 260 articles were manually inspected, with more than 40 identified as highly relevant for our study, demonstrating the growing interest but relatively limited work at the intersection of FL and FMs.

To classify the methods presented, we developed a custom taxonomy based on the stage of the development life cycle—training, customization, and deployment—and further categorized methods based on their complexity, efficiency, and scalability. This classification framework provides clarity on the current landscape of methods and identifies areas where additional research and innovation are needed.

Our analysis emphasized several challenges and practical considerations of integrating FL with FMs. In particular, we highlighted the significance of methods that balance communication efficiency with personalization capabilities, such as selective fine-tuning and additive parameter approaches. These methods are promising for addressing key issues in federated environments, such as data heterogeneity and limited client resources.

We also discussed the application of FMs in the healthcare domain, which remains significantly underserved in terms of adoption within FL settings. This presents substantial opportunities for innovation, especially in developing approaches that handle the complexities of medical data, such as privacy preservation and non-IID data distributions across clients. By exploring healthcare as an illustrative use case, the article highlights the potential for FMs to bring transformative benefits to sensitive domains where data sharing is restricted.

Additionally, our review underscored the need for new research in multiple areas, such as developing or applying novel aggregation techniques beyond the standard FedAVG, exploring in depth privacy-efficiency trade-offs, or studying novel trade-offs in practical applications. This establishes a foundation for future research on integrating FMs with FL, as well as on the practical applications of these methods. First, the presented taxonomy and analysis guide researchers toward areas that require further exploration and innovation. Second, the comparison of techniques assists researchers in developing novel, more efficient, or scalable methods. Last, the study serves as a comprehensive reference and guide to existing literature, facilitating the uptake and familiarization with the literature on this topic.

## ACKNOWLEDGEMENTS

We would like to acknowledge the use of ChatGPT in refining the clarity and readability of this manuscript. The AI assistance was primarily used for tasks such as sentence restructuring, word choice suggestions, and identifying potentially unclear phrasing. We emphasize that the AI was used solely for language enhancement and did not contribute to the generation of research ideas, data analysis, or the interpretation of results. All conclusions drawn and insights presented in this manuscript are solely the product of the authors' own analysis and expertise.

# APPENDIX

This section provides a brief overview of the prerequisite knowledge for FL, FMs, fine-tuning and KD that is used most frequently by the articles discussed in Methods.

## Federated learning

In federated learning (FL), the objective is to collaboratively develop a global model by merging local model updates from participating clients without exchanging the private data stored on their devices. The training process usually involves a central server that orchestrates communication among multiple clients and aggregates local updates into a global model. Each client performs local updates using its own data and transmits these updates to the central server, which then integrates them using various techniques.

In the most basic implementation, during each training round, every client conducts a forward pass using a shared model architecture on its local data and computes an update for all parameters using a shared loss function. The clients then send their model updates to the server, which integrates them to form a global model and sends the updated parameters back to the clients. This iterative process continues, with each client replacing its local model with the global parameters and resuming training as described.

The majority of articles presented in this study use FedAVG to integrate the local client updates, which is defined as:

$$w^{t+1} = \sum_{k=1}^{K} \frac{n_k}{n} w_k^{t+1} \tag{1}$$

where $n_k$ represents the number of samples on client $k$, and $n$ is the total number of samples across all clients. This process weights each client's contribution to the global model according to the size of its local dataset.

## Pre-training foundational models

The main method for pre-training FMs is self-supervised learning (SSL), which involves creating pretext tasks from unlabeled data. This approach eliminates the need for annotated data, allowing to significantly expand the training datasets. Some SSL methods are further extended to incorporate semi-supervised learning, allowing them to leverage any available annotated data. This is particularly useful when the annotations are not related to the downstream tasks or are noisy.

SSL methods can be broadly classified into three categories (i) contrastive learning methods (*e.g.*, *Chen et al., 2020*; *He et al., 2020*), where the pretext task aims to generate similar embeddings for inputs representing the same concepts (*e.g.*, images of the same organ) and distinct (contrastive) embeddings for inputs representing distinct concepts (*e.g.*, prostate *vs.* kidney); (ii) correlation-based methods (*e.g.*, *Bardes, Ponce & LeCun, 2022*), where the pretext task aims to decorrelate embeddings of distinct objects while preserving the variance of embeddings for similar concepts, and (iii) masked input modelling (*e.g.*, *He et al., 2022*; *Baevski et al., 2023*), where the task involves reconstructing original inputs from masked versions of it (*e.g.*, masking words in sentences or patches of images). Recent studies suggest that, despite their differences in methodology and training objectives, SSL

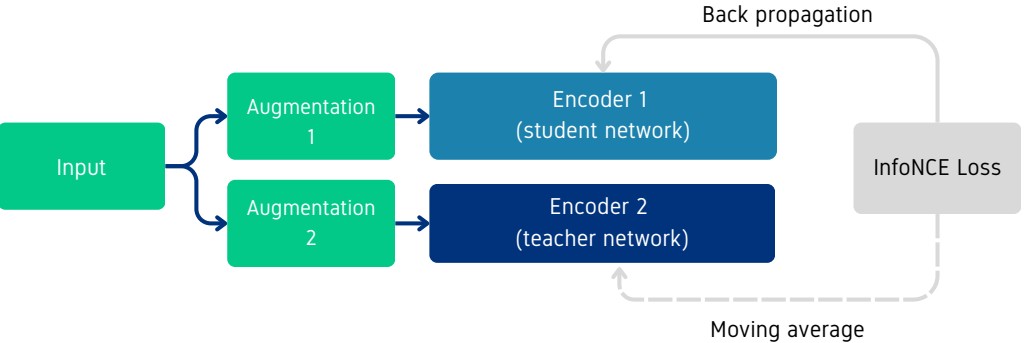

**Figure 7** **Illustration of pre-training FMs using contrastive learning.**

methods are closely related and may produce similar embeddings (*Garrido et al., 2023*). This is because they minimize criteria that are equivalent under certain conditions. These methods are independent of the architecture used, being universally applied to transformer or convolutional based architectures.

The predominant methods used in FL, as discussed in section 'Train', use contrastive learning techniques that involve passing two augmented versions of an input through identical decoders, known as Siamese networks, but updating the decoders differently. The primary decoder, often referred as the student network, is updated through standard back-propagation. The secondary decoder, called the teacher network, is updated using different techniques aimed at preventing the encoders from generating overly the same embeddings, a phenomenon known as feature collapse. These techniques include the use of momentum or EMA as in BYOL (*Grill et al., 2020*; *Cai et al., 2021*). Simpler variants that use a single encoder for both input augmentations are possible but generally less effective *e.g.*, as in SimCLR (*Chen et al., 2020*). The teacher network is usually the only one updated in FL, using global aggregation. An illustration of these techniques is provided in Fig. 7.

In all cases, the loss used is based on the InfoNCE contrastive loss (*Parulekar et al., 2023*).

## Fine-tuning FMs and LoRA

Fine-tuning follows the pre-training the FMs on unlabeled data by continuing the training process using labeled data in a supervised learning manner. This step can be seen as specializing the FMs for a specific task, allowing the model to leverage the general knowledge gained during pre-training and adapt it to particular nuances of the task at hand. While all model parameters can be updated during fine-tuning, resource constraints may induce a more selective approach. One strategy is to update only a subset of the parameters, such as the final layers (also called adapter tuning), which can be effective in projecting and adapting the models' outputs for specific tasks. Alternatively, one can introduce additional, smaller sized parameters at each layer, allowing the model to learn task-specific features while keeping most of the pre-train parameters fixed.

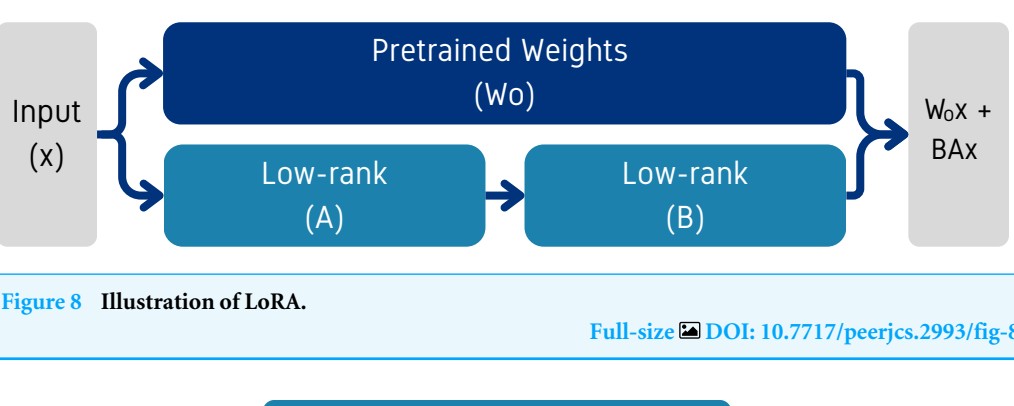

**Figure 8  Illustration of LoRA.**

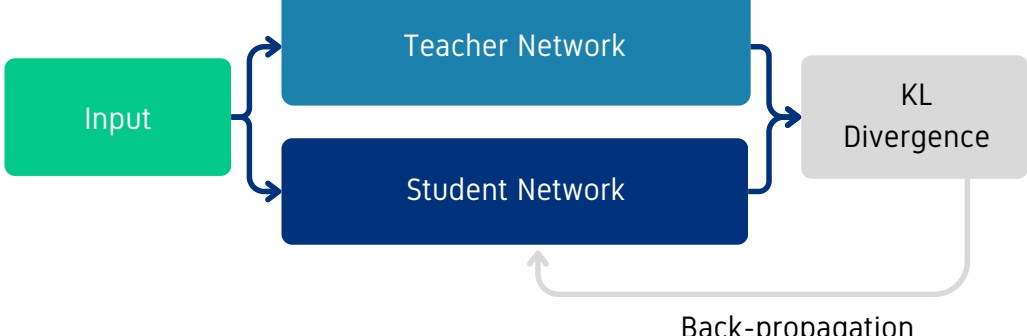

Back-propagation

**Figure 9  Illustration of knowledge distillation.**

The most prevalent method for introducing additional parameters, frequently used in FL fine-tuning is known as LoRA. This method assumes that all model parameters have an intrinsic low rank that can be adapted during fine-tuning. Therefore, each model parameter (*e.g.*, weight matrix) can be decomposed into the sum of the initial parameters and a low rank update expressed as $W_0 x + \Delta W x$. Here $W_0$ represents the frozen initial parameters, and $W x$ is the low-rank update optimized during fine-tuning. This low-rank is defined by two trainable matrices $W x = B A x$, where $A$ encodes the input to a lower dimensional rank and $B$ recovers the output dimension of the original $W_0$. An illustration of LoRA is provided in Fig. 8. In transformer architectures, LoRA is applied to the attention weights rather than the multi-layer perceptronss (MLPs) layers.

LoRA can speed up fine-tuning of large-scale FMs with more than 25%, while decreasing the memory requirements by more than $10^3$. Other LoRA variants, such as quantized LoRA further decrease the memory requirements by up to 90% (*Dettmers et al., 2024*).

### Knowledge distillation

Knowledge distillation (KD) aims to transfer the knowledge from a large FM (called teacher) to a smaller, more efficient model (called student). This process involves training the student model to mimic the behavior of the teacher model by matching its outputs, such as logits or soft targets, rather than relying only on supervised signals such as labels. The most common method for minimizing the distance between the teacher's and student's

**Table 5  List of terms for the queries.** Queries were formed by first combining the first and second terms using the boolean AND operator, and afterwards combining the first, second, and third terms using the same operator.

| First terms | Second terms | Third terms |
|---|---|---|
| Federated Learning | Foundational Models | Healthcare |
| | Self-supervised learning | Medical |
| | Pre-training | Applications |
| | Pre-trained models | Implementation |
| | Fine-tuning | Development |
| | Parameter-efficient fine-tuning | Deployment |
| | Adapter tuning | Inference |
| | Distillation | |
| | Transfer learning | |
| | Compression | |
| | Quantization | |
| | Pruning | |

outputs is the Kullback–Leibler (KL) divergence (*Gou et al., 2021*). In the context of FL, KD is used to transfer knowledge from local models to a global model by minimizing the divergence between their outputs. This approach allows the global model to benefit from the collective knowledge of the local models. An illustration is provided in Fig. 9.

### List of terms for literature search

The complete list of terms used to develop queries for the literature review is illustrated in Table 5.

### Funding

This research was funded by the European Union Horizon Europe R&I program through the project FLUTE (Grant agreement no.: 101095382). The funders had no role in study design, data collection and analysis, decision to publish, or preparation of the manuscript.

### Grant Disclosures

The following grant information was disclosed by the authors:
The European Union Horizon Europe R&I program through the project FLUTE: 101095382.

### Competing Interests

Cosmin-Andrei Hatfaludi and Alex Serban are employees of Siemens SRL, Foundational Technologies, Brasov, Romania.

## Author Contributions

- Cosmin-Andrei Hatfaludi conceived and designed the experiments, performed the experiments, analyzed the data, performed the computation work, prepared figures and/or tables, authored or reviewed drafts of the article, and approved the final draft.
- Alex Serban conceived and designed the experiments, performed the experiments, analyzed the data, performed the computation work, prepared figures and/or tables, authored or reviewed drafts of the article, and approved the final draft.

## Data Availability

This is a literature review.

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
