# Peer review of "Foundational models and federated learning: survey, taxonomy, challenges and practical insights"

_PeerJ Computer Science, doi:10.7717/peerj-cs.2993_

## Round 0.1 · original submission · Major Revisions

Please consider the reviewers' comments and clarify or modify accordingly with them.

Reviewer 1 ·

Basic reporting

Seems okay.

Experimental design

Good plan.

Validity of the findings

Not sure. Can be rechecked.

Additional comments

N/A

Annotated reviews are not available for download in order to protect the identity of reviewers who chose to remain anonymous.

Reviewer 2 ·

Basic reporting

1. Abstract should contain the reason of this study
2. The survey and review are two different terms with different meaning. authors should stick to one.
3. The introduction is somewhat cluttered with background information and technical details. It would benefit from a clearer separation between the general introduction to the topic and the specific objectives of the paper
4. Add comparison table in the related work section to highlight the difference between this review and existing reviews

Experimental design

1. include the inclusion, exclusion criteria of the papers
2. authors should add studies from 2025 as well
3. Protocol to review the study is missing
4. One of the main aim of the review paper is to provide comprehensive challenges and open directions. which are missing in this paper.
5. main sources of the selected studies should be presented in tabular form.

Validity of the findings

1. authors should highlight the main findings of the study in discussion section with the help of graphs and other visuals.
2. Outline study limitations
3. Add a graph that illustrate the trend of the techniques over the years

---

## Round 0.2 · accepted · Accept

Your manuscript is ready for publication.